# Program sustainability post PEPFAR direct service support in the Western Cape, South Africa

**Jessica Chiliza**[1]*, **Richard Laing**[1,2], **Frank Goodrich Feeley III**[1], **Christina P. C. Borba**[3]

**1** Department of Global Health, School of Public Health, Boston University, Boston, Massachusetts, United States of America, **2** School of Public Health, University of Western Cape, Bellville, South Africa, **3** Department of Psychiatry, Boston University School of Medicine, Boston Medical Center, Boston, Massachusetts, United States of America

* jchiliza@bu.edu

**Data Availability Statement:** All relevant data are within the paper and its Supporting Information files.

**Funding:** The author(s) received no specific funding for this work.

## Abstract

### Background

Public health practitioners have little guidance around how to plan for the sustainability of donor sponsored programs after the donor withdraws. The literature is broad and provides no consensus on a definition of sustainability. This study used a mixed-methods methodology to assess program sustainability factors to inform donor-funded programs.

### Methods

This study examined 61 health facilities in the Western Cape, South Africa, supported by four PEPFAR-funded non-governmental organizations from 2007 to 2012. Retention in care (RIC) was used to determine health facility performance. Sustainability was measured by comparing RIC during PEPFAR direct service (20072012), to RIC in the post PEPFAR period (2013 to 2015). Forty-three semi-structured in-depth interviews were conducted with key informants. The qualitative data were used to examine how predictor variables were operationalized at a health facility and NGO level.

### Results

Our qualitative results suggest the following lessons for the sustainability of future programs:

- Sufficient and stable resources (i.e., financial, human resources, technical expertise, equipment, physical space)

- Investment in organizations that understand the local context and have strong relationships with local government.

- Strong leadership at a health facility level

- Joint planning/coordination and formalized skill transfer

**Competing interests:** The authors have declared that no competing interests exist.

- Local positive perceived value of the program

- Partnerships

## Conclusion

Sustainability is complex, context dependent, and is reliant on various processes and outcomes. This study suggests additional health facility and community level staff should be employed in the health system to ensure RIC sustainability. Sustainability requires joint donor coordination with experienced local organizations with strong managers before during and after program implementation. If the program is as large as the South African HIV effort some dedicated additional resources in the long term would be required.

## Introduction

Over the last twenty years, new sources of donor funding from private foundations, philanthropists and the private sector have significantly expanded the field of HIV/AIDS care. Global funding for HIV increased annually from $1.2 billion in 2002 to $8.6 billion in 2014 though there was a significant plateau of global HIV funding, following 2008, due to the global financial crisis [1]. The increased funding resulted in a small decrease in the incidence of HIV globally. At the same time the increasing number of PLHIV necessitated [2] that low and middle income countries (LMIC) augment their domestic HIV programs. In 2012 UNAIDS reported the main source of global HIV funding came from domestic resources [3]. Additionally in 2012 the, World Banks' re-classification of country income levels [4] had a negative influence on the flow of donor funding, especially with the delineation of middle income countries (MICs) into lower and upper. Their criteria has been critiqued for being based on aggregate income levels, rather than social inequality [5].

Due to these changes in global donor funding there has been increased interest in transitions or graduations, when large donor funded programs decrease funding or exit a country, requiring the local government to take financial responsibility for their health programs. Transitions have been described as a "new art," [6] which is "complex" [7].

Recently additional research has emerged on PEPFAR, the Global Fund and other donor transitions globally. PEPFAR transition literature from South Africa, Nigeria and Uganda have highlighted decreased access and reduced quality of care, preventative and community outreach services and retention in care after the withdraw of PEPFAR funding [7–10]. An evaluation in Nigeria found, post transition, a decrease in access to laboratory services which affected viral load testing (92% to 64%; p = 0.02), staff shortages due to a lack of incentives to retain staff (80% to 20%; p<0.01), and reductions in tracing systems for HIV patients (100% to 44%; p<0.01) and community testing services (84% to 64%; p<0.01) [7].

### Program sustainability

The central premise of transition directly relates to sustainability and the long-lasting effects of donor funds. How to nurture the continuation of effective program benefits, especially after donors leave, should be a priority for the public health community just as much as implementing new programs. If efforts to scale up and sustain effective health investments are not prioritized, donors are constantly re-inventing the wheel, wasting scarce resources and time [11–

13]. Also, there is a moral imperative to sustain programs that are effective. This is particularly true for chronic diseases such as HIV/AIDS.

Very little is known about what happens to programs or their outcomes when donor funding terminates. The literature estimates at least 40–50% of social programs collapse within a year after funding ends [14, 15]. Additionally, Cekan found that very few (1%) development projects are evaluated post donor funding [16].

The research on sustainability is broad, and the quality of the research methods used is generally poor. There is no clear agreement on a definition, little analysis on sustaining programs in a complex health system, and only a handful of lessons learned about large donor transitions have been reported. Local governments are left to sustain donor instituted programs as best they can or to let them expire from lack of funding or attention [6]. Wickremasinghe et al. [17] highlight that to achieve country ownership, strong relationships and engagement with government, in the design, implementation and evaluation are key. Ultimately, new programs need to be embedded within the local health system for government to adopt them [17].

Most donors set the program priorities and control the rules of the donor/grantee playing field, which includes defining sustainability. From the early days of international health Pan-American Health Organization (PAHO) and Rockefeller Foundation have equated sustainability with financial sustainability [18]. More recently, PEPFAR's HIV/AIDS Sustainability Index and Dashboard (SID) focuses mainly on national level policies and financial sustainability [19].

There have been various international program design initiatives to increase the effectiveness and sustainability of international aid. In the mid-1990's, the sector wide approach (SWAp), was introduced into international development circles. SWAp was a mechanism specifically targeted at health initiatives which intended to shift the decision making of the health budgets back to host governments, instead of international health donors. Donor funds would be put in a common fund, and local government would coordinate, plan, monitor the budget all health funding based on local priorities [20]. In theory, this would be more cost effective, increase sustainability and reduce duplication by donors and host governments [21]. In practice, countries implemented SWAp differently which made it difficult to measure, additionally there were other strategies introduced to increase the effectiveness of aid. To date these strategies, include the Paris Declaration and Accra Agenda for Action and Busan Partnership for Effective Develop of Co-operation of to improve the coordination of aid effectiveness, have shown few tangible effects on health outcomes [22]. The US government and Global Fund opt-ed out of SWAp, while increasing disease specific funding, directed at non-governmental organizations (NGOs). Research from Uganda [20], Mozambique [23] concluded SWAp received a small percentage of health funding, as PEPFAR and the Global Fund funding dramatically took over the international health funding scene.

The donor community has equated sustainability with financial capacity. Though consistent financial support is a key component of sustainability, we would argue along with others [6, 24, 25] this definition needs refinement. It is important to understand program sustainability to ensure that scarce health system resources, in addition to funding, are effectively used.

## Transition in South Africa

South Africa is the country with the greatest number of people living with HIV globally (7.5 million) and with 4.1 million adults on treatment, [26] South Africa is running the largest HIV treatment program globally [10]. From 2004 to 2018, the United States President's Emergency Fund for AIDS Relief (PEPFAR) invested $5.9 billion into the South African HIV/AIDS response [27]. Most PEPFAR funds in South Africa were distributed to NGOs that supported

state health facilities to strengthen HIV/AIDS care and treatment programs. During the initial stages of PEPFAR, the majority of funds supported the distribution of antiretroviral treatment (ART) [28]. Over the years, there have been various changes to PEPFAR's leadership and strategy in South Africa. In 2012, there was a planned transition from service delivery to health systems strengthening, a gradual budget decrease and handover of the HIV program to the South African government (SAG). At this time, a study in Durban, South Africa estimated that 20% of clients were lost to follow-up by care and treatment programs in South Africa [29] mainly due to the poor treatment in government health facilities[8]. Based on Cloete's estimate, [30] Kavanagh approximates the PEPFAR transition affected 50,000 to 200,000 people living with HIV (PLHIV) [10]. One of the main critiques was that the PEPFAR transition focused solely on care and treatment with no plans for other PEPFAR funded activities (i.e. prevention). Others found that at the national level there was a lack of PEPFAR leadership and a lack of clear guidance and communication around the pace of the budget decrease [31].

This high loss of clients, resulting in a lack of adherence to treatment regimens and consequent possible increase of drug resistant strains of the virus was a major concern. No formal evaluation of the PEPFAR transition in South Africa was ever undertaken; therefore, it is unclear what happened to the thousands of clients on treatment and to staff, and NGOs formerly funded by PEPFAR, or to HIV outcomes, such as ART retention and mortality.

## Western Cape transition

The Western Cape is distinct from other South African provinces. With a concentration of tertiary health services and prominent internationally renowned HIV experts, it has some of the best health outcomes in South Africa [32, 33]. Compared to other provinces, the Western Cape also has the lowest HIV prevalence at 7.8% (2012). Historically, the Western Cape has the oldest health system focused on white urban populations, and governed by strong leadership [34]. Governed under the political opposition party, the AIDS program, specifically the PMTCT program was the first of its' kind in South Africa [35].

Partially due to the availability of resources and strong leadership to make critical decisions and provide guidance, the Western Cape Government Health took the initiative to begin the PEPFAR transition process earlier than other provinces [36]. Over the course of two years (2011–2012) a memorandum of understanding was developed, a detailed database was created, staff cadres and salaries were aligned to government staffing norms and policies and hospitals and district staff were consulted. This process resulted in 40% (n = 78) of PEPFAR clinical and administrative posts being absorbed by government or 13% of all the Western Cape PEPFAR posts [36]. This paper aims to assess how the PEPFAR program in the Western Cape province of South Africa withdrew to identify factors associated with sustained performance.

## Methods

All participants provided written informed consent. Ethical approval was received from Boston University Medical Campus Institutional Review Board (Protocol Number: H-37238) and the Biomedical Science Research Ethics Committee, University of the Western Cape, Cape Town, South Africa (BM18/5/2).

We evaluated PEPFAR program outcomes that were sustained following the withdrawal of funding for direct service support (2007–2012) and the factors that led to program sustainability. Health facilities were characterized by their ability to sustain HIV program outcomes post PEPFAR funding for direct service support and the organizational (i.e. health facility and NGO), programmatic, and contextual factors that led to sustainability were analyzed.

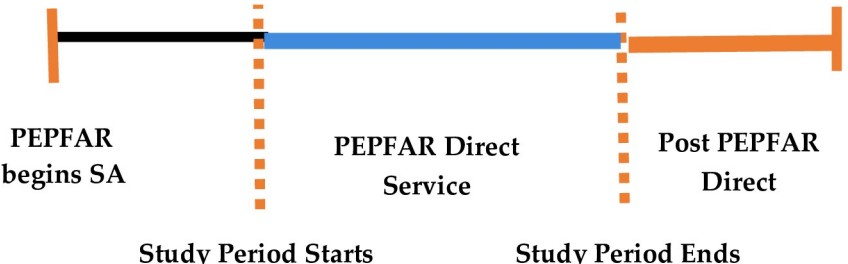

**Fig 1. PEPFAR strategy timeline South Africa.**

A mixed-methods approach was used to examine health facilities supported by four local PEPFAR treatment NGOs from 2007 to 2012. This paper reports on the qualitative results of the study. Quantitative results will be reported elsewhere. Financial sustainability is a key element to achieving program sustainability, but it is not the only factor, therefore the focus of this study was on the non-financial characteristics of sustainability.

PEPFAR intended to terminate direct service support in 2012/2013 in South Africa. This study used this planned direct service end date as the break point of our analysis. We refer to the period during direct service (2007–2012) as, "PEPFAR direct service" and after direct service (2013–2015) as "post PEPFAR direct service" (Fig 1). Retention in Care (RIC) was used to measure health facility performance. RIC and mortality are key indicators that demonstrate the long term sustainability of the ART program [37]. The study used the same definition the Western Cape Government Health (WCGH) uses for RIC which is: patients on first line treatment + second line treatment + third line treatment + patients who stopped ART, divided by (total number of patients on treatment–total transferred out). Sustainability was measured by comparing RIC during PEPFAR direct service 2007 to 2012, to RIC in the post PEPFAR period 2013 to 2015.

RIC is key to achieving the global 90-90-90 U.N goals: 90% of all people living with HIV will know their HIV status, 90% of people with diagnosed HIV infection will receive sustained ART and 90% of all people receiving ART are virally suppressed. RIC is currently used as the main indicator to achieve the second 90: 90% of people with diagnosed HIV infection will receive sustained ART [38]. The 90-90-90 goals have also coincided with the leveling of donor funding globally and the "transition" away from large global donors [39].

## Study sample

This study examined health facilities supported by four local PEPFAR treatment NGOs from 2007–2012: (1) Kheth'impilo (KI), (2) Anova Health Institute (3) Right to Care, (4) TB, HIV/ AIDS, Treatment Support and Integrated Therapy (that'sit). Since the interviews did not ask subjects about personal health details or collect protected health information, this study was declared exempt from human subjects review by Boston University Medical Campus Institutional Review Board and the Biomedical Science Research Ethics Committee at the University of the Western Cape. Informed consent was received from each participant.

These NGOs were the four PEPFAR supported HIV care and treatment organizations working in the Western Cape from 2007 to 2012. Right to Care's timelines were slightly later, from 2009 to 2015. Each of the NGOs worked in a specific geographic region. The four NGOs under study supported 100 primary health care facilities in the Western Cape with PEPFAR funds between 2007 and 2012. A description of the health facilities and their outcomes can be

Table 1. Summary of qualitative sample by health facility characteristics and outcomes.

| Facility Characteristics | | | Facility Outcomes | | |
|---|---|---|---|---|---|
| Geography | Urban | 9 (40.9%) | Overall RIC (2007–2015) | Low (<59.9%) | 12 (54.5%) |
| | Rural | 13 (59%) | | High (>60%) | 10 (45.5% |
| ART Patient Volume | Low (≤ 700) | 8 (36.4%) | Sustainability | Poor (< -5.0%) | 14 (63.6%) |
| | Medium (700.9 ≤ 2999.9) | 6 (27.3%) | | Sustained (-4.9% to 4.9%) | 7 (31.8%) |
| | High (>3,000) | 8 (36.4%) | | Improved (>5.0%) | 1 (4.5%) |
| Government ownership | CoCT | 6 (27.3%) | | | |
| | WCGH | 15 (68.2%) | | | |
| | Combined | 1 (4.5%) | | | |
| NGO Support | Anova | 5 (22.7%) | | | |
| | Right to Care | 4 (18.2%) | | | |
| | Kheth'impilo | 5 (22.7%) | | | |
| | that'sit | 5 (22.7%) | | | |
| | Kheth'impilo/ Anova | 3 (13.6%) | | | |

found in Table 1. This study excluded tertiary and district hospitals due to the history of the PEPFAR program, which began in tertiary facilities and offered patients access to HIV specialists. Due to these differences, we only included facilities from a PHC (Primary Health Care) level.

The quantitative data was used to select the qualitative sample. Purposive sampling was used not to select a representative sample, but a broad distribution of health facilities to examine different contexts and better understand sustainability factors. Six health facility characteristics (geographic area, PEPFAR NGO, ART patient volume, government ownership, sustainability of retention in care (RIC) and RIC at 12 and 24 months) were used to investigate sustainability.

## Data collection

In total, 43 in-depth interviews were conducted across a five-month period (October 28, 2018 to April 3, 2019). The interviews were conducted with health facility managers from 20 primary health care facilities, one clinical nurse practitioner who is a high-level practitioner just below a doctor and one lower-level staff nurse (Table 2). Fourteen key informant interviews were conducted with eight government officials and five NGO program managers (Two participants were interviewed from one NGO.). To gain clarity and a better understanding of the themes in the first set of interviews, a second round of follow up interviews were conducted with eight existing study participants.

To guide the semi-structured in-depth interviews, interview guides and information sheets were developed. Interview guides were validated with two health facility managers in Kwa-Zulu-Natal. Three different interview guides were developed specific to each participant category (Table 2).

## Data analysis

The interview recordings were transcribed, coded and themes were identified using a grounded theory and a thematic analysis using Nvivo 12 Pro [40]. Thematic analysis allows for theories to emerge from the data without trying to fit "preconceived ideas and theories" into the data, grounding the analysis in the data. This inductive process allowed for the observation of repeated patterns to allow for theories to emerge organically from the data. The

**Table 2. In-depth interviews.**

| 5 | NGO |
|---|---|
| | 4 x NGO Program Directors |
| | 1 x NGO Provincial Assistant Manager |
| 22 | Health Facility |
| | 20 x Health Facility Manager or Operational Manager |
| | 1 x Clinical Nurse Practitioners Nurses |
| | 1x Staff Nurse |
| 8 | Government |
| | 6 x Provincial Government Officials |
| | 2 x District Government Officials |
| Total First Interviews: 35 | |
| Second Interviews | |
| 3 | NGO |
| 2 | Provincial Government |
| 3 | Health Facility Managers |
| Second Interviews Total: 8 | |
| Grand Total: 43 | |

sustainability factors guided the themes to be analyzed, but the analysis allowed for additional themes to emerge. The analyses used a flexible analytic approach which allows the investigator to move back and forth between the data and analysis to connect emerging themes [41, 42].

# Results

Our qualitative results found nine key factors that lead to program sustainability. The factors associated with sustainability focus on people (e.g. health facility leadership, skilled staff, stable human resources, perceived value), relationships (long standing presence, partnerships) systems (donor coordination and formalized skills transfer) and additional resources (financial, human resources, technical expertise, equipment, physical space). These factors were integrated with the broader transition and sustainability literature to produce a framework to maximize program sustainability outlined below (Tables 3–8).

## Joint planning

The donor and local government at every level of government (health facility, district, and provincial level) and NGO need to plan together throughout the life of the donor funded program.

**Table 3. Prior to program launch.**

| Actions: National Level | |
|---|---|
| Donor | Before a funding announcement is put out donors need to work with national and provincial level stakeholders (i.e Ministry of Health, National Treasury, AIDS Councils) to understand local needs and gaps. |
| Grantee | Provincial government to work with facilities and communities to understand local needs. |
| Donor | Prioritize funding, local gaps and innovation. |
| Donor | Prioritize funding organizations that have a record of accomplishment in the geographical area. |
| Donor/ Grantee | High level commitment |
| Grantee | Ideally established donor coordination system which communicates with all levels of governments |

**Table 4. Beginning of the program.**

| Provincial Level | |
|---|---|
| **Donor** | Respect the needs and opinions of the grantee. |
| **Donor** | Has the skills to fill the needs of the grantee. |
| **Donor/Grantee** | Recognize it will take extra time to coordinate donor funds. |
| **Donor/Grantee** | Understand the importance of human resource stability since it affects outcomes |
| **Donor/Grantee** | Transparency of program activities and resources including budgets. |
| **Grantee** | Needs local champions to keep motivation high. |
| **Grantee** | Characteristics of leader who is based at the lowest donor/grantee interface<br>• Takes ownership of program staff and communicates clear roles and responsibilities<br>• Empathy for patients and staff<br>• Creates strong teams<br>• Able to motivate/incentivize<br>• Uses data to make decisions<br>• Plans for the future<br>• Good communication skills<br>• Understands needs of facility |

The main goal of the donor should be to fill needed gaps and let government lead the planning process. To ensure the planning process is authentic, the donor needs to have the skills to fill gaps, while respecting the local governments needs. Additionally, it helps if local government understands their own health system needs and has an established donor coordination system at a provincial level. Funding for a liaison person at the provincial level to coordinate transition activities was highlighted as a key to sustainability. This person would be responsible for ensuring transparency of donor funded activities and work with government to ensure the program is integrated into the local health system.

## Long-standing presence

When an NGO has an established office in the geographic region, they understand the context, local policy and have strong relationships with government, which builds trust and results in more sustainable outcomes. These relationships and trust led to post donor funding opportunities for formerly funded PEPFAR NGOs. The most important PEPFAR outcome the

**Table 5. Continued: Beginning of the program.**

| Provincial Level | |
|---|---|
| **Donor/ Grantee** | Consultation with a wide range of stakeholders to coordinate donor funded program (i.e. Provincial treasury, civil society, leaders from provincial, district, sub-district and health facility, HIV activists). |
| **Donor/ Grantee** | Donor-funded liaisons are placed in national and provincial offices to assist with program implementation and coordination. |
| **Donor/ Grantee** | Develop a program roadmap with clear timelines. Define and communicate overall goals, outcomes and coordination processes of donor-funded program. |
| **Donor/ Grantee** | Develop a program implementation plan with all stakeholders. Define sustainability requirements. Not every activity must be sustained.<br>• Align donor salaries with local salaries<br>• Cost the program<br>• Prioritize the funding of extra resources and human resources in smaller clinics.<br>• Consider program beneficiaries and transience of different types of staff.<br>• Consider the importance of community health workers and administrative support |
| **Donor/ Grantee** | Develop an M&E plan for the program |
| | Align donor program indicators and staffing structures with local system. |

**Table 6. Mid-term.**

| Provincial Level | |
|---|---|
| **Donor/ Grantee** | All stakeholders discuss policy, budget, program, donor, local contextual changes and challenges facing the program |
| **Donor/ Grantee** | Look for ways to create partnerships between government and/NGO, and between NGOs. |
| **Donor/ Grantee** | Continuation of coordination meetings with grantee at lowest grantee/donor interface<br>• NGO and health facility<br>• Donor-funded staff and local staff<br>• Provincial level government officials<br>• HIV/AIDS activists and community leaders |

WCGH wanted to sustain was the transfer of skills. We found this transfer must be formalized at both a centralized (i.e. provincial level) and decentralized level (i.e. health facility level). The centralized level of government should decide the human resources that are essential, which should be followed up with adequate financial resources. At a de-centralized level, a strong health facility manager is required to ensure the skills of donor funded staff person are transferred to local staff to sustain skill sets. One health facility manager ensured six months before the PEPFAR staff member left, they mentored and trained a local staff member in their job responsibilities.

## Partnerships

One of the main factors, which led to sustained infrastructure, resources and improved donor coordination was due to donor/grantee partnerships. The donor and grantee were committed to providing resources toward a common goal—controlling the HIV epidemic. This commitment played out in several ways. In some instances, the PEPFAR NGO built a pharmacy and

**Table 7. Transition period (final 2–5 years before program transitioned).**

| National Level | |
|---|---|
| **Donor/ Grantee** | Official transition plan developed by consultants with input from a wide range of stakeholders and funded by the donor |
| | • High level plan<br>• Implementation plan |
| **Donor/ Grantee** | Political commitment to the transition, which includes a financial commitment. |
| **Provincial Level** | |
| **Donor/ Grantee** | Grantee leads review process of program outputs and outcomes to assess program effectiveness |
| **Donor/ Grantee** | If patients are moving from NGO care to the public system, develop a tracking system to monitor progress. |
| **Donor/ Grantee** | • Formalize the skills transfer, which should be coordinated at centralized and decentralized levels.<br>• Prepare the public health system to absorb donor funded activities and staff |
| **Donor/ Grantee** | Review M&E data. |
| **Donor/ Grantee** | Clear communication with all stakeholders regarding M&E updates and budget timelines |
| **Grantee** | Develop local transition plan. |
| | • Stakeholders should decide what they can realistically sustain within their budgets.<br>• Review all donor funded activities<br>• Possibility to use a staggered approach to absorb donor- funded resources |
| **Donor** | Provide capacity and technical assistance where needed |

**Table 8. Post transition (3–5 years after the end of the program).**

| Provincial and National Level | |
| --- | --- |
| **Donor/Grantee** | • Monitor sustainability indicators |
| | • Conduct post-evaluation of program using a time series analysis |
| | • Share insights and lessons learned with all stakeholders |

the local government created pharmacy posts to manage the pharmacy. In another example, a medical cart was funded by the PEPFAR NGO and commodities were stocked by local government.

> "I know it was not so difficult to get things (PEPFAR direct service) but now (post PEPFAR) it's difficult. You need to write a motivation first to get a table or chair and say there is an underspending."(NGO Program Director, Rural)

Local government understood their service delivery gaps and required PEPFAR expertise post PEPFAR direct service support. Valued and trusted for their expertise, the district hired a former PEPFAR funded medical doctor to provide Nurse Initiated Management of Anti-retroviral Therapy (NIMART)mentorship and an NGO to assist with facility management trainings.

> "I think, look, before it was never just about people coming in and doing the work for us. There was that transferring of skills. There was an ongoing process. There were relationships being built, when the mentors, from the different NGOs would come in, they would not just focus on their teams. They would look at; they would work with the (local government) team." (Provincial Government Official, Urban)

## Human resources

In the Western Cape (unlike some other areas of South Africa), PEPFAR was able to plan and formalize the transition of human resource posts from PEPFAR to local government. The retention of these posts and the relationships created led to a greater number of skilled staff being retained in the local health system. The PEPFAR funded NGO found investing in lower-level cadre of health facility staff (i.e. nurses, data capturers) was more sustainable, because they were from the local communities they were working in and less likely to leave the geographic area. Additionally, human resource stability was key to sustaining health facility outcomes. A government official mentioned they witnessed a decrease in health facility outcomes when there was human resource instability, when PEPFAR changed strategies to health systems strengthening support.

> "It's not that I've read any evidence on this but just what we pick up in the system. You see that slump (in the data) and you see people pick it up and pull it together and move forward.". . ..."when you pull it (donor funding) out you will see a dip, but at some point the team that stays behind develops a sense of resilience."(Provincial Health Official, Urban)

## Strong facility manager

Sustainability was the result of dedication and the extra time that health facility leaders invested in coordinating donor funded activities at a facility level. A strong health facility manager needed to have a number of characteristics including time management and organizational skills, is motivated, has good communication with donor and local staff, able to manage

stress, as well as plan for the future. Part of the role of the health facility manager included taking ownership of donor staff to ensure they are used effectively. High performing/high sustainability health facility managers did not wait for PEPFAR NGOs to define their staff members' roles, they placed PEPFAR funded staff into gaps, integrated them into staff meetings and fired those who were too challenging.

> "I always believe that if the staff [emphasis] are happy you get more out of them, than when they are not, so when you go into a facility, you look at first your staffing issues before you actually look at the patient issues, 'cause patient issues can always sort but once you sorted your staffing problems and when they are seen as problems, you can sort that out and they are willing to sort out your patient issues for you."(Health Facility Manager, Urban)

We observed there was more motivation by local health facility mangers to take ownership of the PEPFAR program when they felt the donor program was beneficial to them and the communities they served. Therefore, perceived value by the grantee was a factor that led to program outcome sustainability.

> "And it also improved your other services because although they were only employed for ARVs, I trained them in Integrated Management of Childhood Illness(IMCI). Yeah so and that, that was one of the things I always did with the (PEPFAR) NGOs is that the person that's employed with us but remember I'm going to train them in the capacity of them, so that every patient, they will see to all the other needs of the patient." (Health Facility Manager, Urban)

### Additional resources

Many of the top performing health facilities were provided with additional PEPFAR NGO support, including equipment (e.g. lactose meters, scales, computers) and infrastructure (e.g. extra counselling rooms, gardens) and with multiple PEPFAR posts from the human resource transition.

> "But look, bottom line is, there is more that can be done with resources and hands-on deck. Definitely!" (Provincial Government Official, Urban)

### Discussion

Our results add useful new insights to the current broad transition and sustainability literature. The planning and transition of donor funded activities should be led by the local government at a centralized level: either the provincial or district level. Congruent with the Avahan studies in India, the institutionalization of specific donor program components—mainly budgets, reporting systems and staff structures—are important from the beginning of the program [43]. Effective aid, and sustainability are reliant on alignment with the countries health and development priorities [44]. It is important for government and the NGO to be included into the design and implementation of the program since they will ultimately own the program when the donor pulls away [45]. Beracochea succinctly states, "Effective aid is by design, not by default"[45].

While the literature highlights the importance of leadership, our study specifies the qualities which a leader should display. Long tenure does not equate with leadership. Although we

found that health facility managers stayed in their position for an average of 13 years, and on average had worked for local government for 23 years, long tenure was no guarantee of strong leadership and management skills. The best performing health facilities (high RIC and sustainability scores) were led by a health facility manager who had an understanding of the health facility's needs, set clear roles and responsibilities with donor and local staff, had empathy for patients and cared for the wellbeing of their staff, while creating a strong motivated teams.

There are contextual factors that likely assisted sustainability in this study. This study was based in a province that is politically and financially stable. Additionally, PEPFAR funded an abundance of HIV research, which assisted the grantee and donor in understanding the HIV epidemic in the Western Cape. A better understanding of context will assist large donors when transitioning programs to local government to help ensure program gains are retained post donor funding. This study period also coincided with a time of change in South Africa, when the political support for ART access increased, national HIV budgets increased substantially, AIDS activism was noteworthy and when task-shifting and ART treatment guidelines became more receptive to placing more patients on treatment. These positive changes at a local level supported PEPFAR program's goals, and the sustainability of the HIV program.

PEPFAR has been criticized for its vertical or single disease approach that tends to weaken the capacity of the local health system [46]. Some research has shown that vertical programs can improve health outcomes [47, 48], although the impact of vertical programs are not as effective as developing local policy or improving the local health system [49]. Others have argued that vertical programs displace funds from other more significant disease burdens [24, 50], increase the brain drain from the public system to donor NGOs who tend to pay higher salaries [24], and adds more work to overworked health care workers. The main concern is that if the public-sector workforce and infrastructure are undermined many African countries they will not reach their 90-90-90 goals [51]. In 2005, the Mozambique Ministry of Health and Health Alliance International took a health systems strengthening approaching, integrating ART services into the existing primary health care system. This "diagonal" approached proved successful, increasing HIV testing rates, reducing loss to follow up and expand HIV services geographically [52]. This approach also strengthened the PHC system, including laboratory and pharmacy services.

Though vertical programs have been criticized for creating parallel health systems, we found that the vertical PEPFAR support was not necessarily a barrier to sustained outcomes in the control of a priority disease. The lesson for future donors is the need to integrate their programs into existing local health structures for program outcomes to be sustainable. Practically this means local governments can place donor-funded vertical program staff into the health system but ensure HIV testing referrals and lab services are integrated in the public health system. The manager at the donor/grantee interface needs to take ownership of the donor program to ensure the donor support is streamlined and efficient for all facility staff and patients.

Donors and local government need to jointly create a sustainability or a phase out plan for every donor-funded activity. It is important to note that not every program activity needs to be sustained. The key question to ask is, *Is the sustainability of the outcomes relevant to the objectives of the intervention or activity*?[53]. Donors need to be especially careful about phasing out human resources in smaller health facilities that will struggle to maintain program outcomes because they are usually absorbed into other services in the health facility. "*We found post PEPFAR direct service, larger health facilities could allow former PEPFAR trained staff to continue to work in the HIV program or were able to sustain PEPFAR's vertical approach, while in smaller health facilities, PEPFAR trained staff were absorbed into other health services since there were fewer staff.*" Strong leaders at the lowest level of the grantee (i.e health facility), plus retention

of community health workers, administrators and data capturers were key to ensuring that positive health outcomes were not lost post donor funding.

## Conclusion

The results of this sustainability study provide concrete guidance for donors, NGO's, philanthropists, and local governments about how to channel donor funding to improve health outcomes. The results of this research can be integrated into program plans to maximize the sustainability of program outcomes. These policy recommendations set the sustainability factors within the context of transition to provide further guidance for donor transitions. To ensure the sustainability of outcomes of future transitions, the PEPFAR transition should have been formally evaluated by PEPFAR to ensure learnings could be applied to other countries going through a similar process.

The Western Cape PEPFAR program was able to transfer and sustain skilled health facility workers via the formal transition, sustain HIV expertise, maintain infrastructure and ensure a strong HIV program. In part, this was due to the strong and stable leadership in the province, formalized skill transfer at a centralized and de-centralized levels, and an abundance of HIV research on the Western Cape. Donors also need to be careful when phasing out human resources in small health facilities, because their specialized skill set will be lost then they are used in other areas of the facility. Though not the focus of this study, the ability of the local government to finance the majority of the HIV program budget was one of the key sustainability components. While research and HIV expertise were not initially defined as sustainability factors, the deep understanding if the Western Cape HIV epidemic and support in policy forums by HIV experts, played a significant role in building a strong HIV program.

This study was unable to identify a single predictor of sustainability. This was not surprising as sustainability is complex, dependent on the context, and relies on various processes and outcomes. What was clear is that future disease specific donor funded programs need to be intentionally integrated into health systems or use a diagonal program approach. If global efforts are going to make progress towards the 90-90-90 HIV goals, donors and local governments need to strategically plan for sustainability from the beginning of any donor funded program, while integrating external investments within local health programs and structures. While we have the tools to end the HIV/AIDS epidemic, global funding that would have supported the 90-90-90 goals has been withdrawn, which has undermined these efforts.

Our study outcomes can be generalized in planning for program sustainability. The following tables (Tables 3–8) provide a checklist for donors and grantees at each phase of a program.

## Supporting information

**S1 File. Health facility interview guide.**
(DOCX)

## Acknowledgments

I would like to thank my DrPH committee Richard Laing, Frank Feeley, Alana Brennan, and Debra Jackson for their tireless support and commitment to this research. Thank you to Christina Borba for your training in qualitative research methods and to the BU students Kate Riffenburg and Laura Tabbaa. A huge thank you to all of the health facility managers, PEPFAR NGO managers and City of Cape Town and Western Cape Government Healthofficials for taking the time to be interviewed.

## Author Contributions

**Conceptualization:** Jessica Chiliza.

**Data curation:** Jessica Chiliza.

**Formal analysis:** Jessica Chiliza.

**Investigation:** Jessica Chiliza.

**Methodology:** Jessica Chiliza, Richard Laing, Frank Goodrich Feeley III, Christina P. C. Borba.

**Project administration:** Jessica Chiliza.

**Software:** Jessica Chiliza, Christina P. C. Borba.

**Writing – original draft:** Jessica Chiliza.

**Writing – review & editing:** Jessica Chiliza, Richard Laing, Frank Goodrich Feeley III, Christina P. C. Borba.

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
