## [Decision Letter · Decision Letter 0]

6 Jan 2021

PONE-D-20-37215

Program Sustainability Post PEPFAR Direct Service Support in the

Western Cape, South Africa

PLOS ONE

Dear Dr. Chiliza-

Thank you for submitting your manuscript to PLOS ONE. After careful consideration, we
feel that it has merit but does not fully meet PLOS ONE’s publication criteria as it
currently stands. Therefore, we invite you to submit a revised version of the
manuscript that addresses the points raised during the review process.   I have
carefully reviewed the manuscript and the reviewer comments, and please provide a
major revision, taking the comments into account, and and resubmit your manuscript
for reconsideration. Please note all reviewers felt this is an important and timely
topic.  Please take the reviewer comments into account and set out your responses to
us in detail. Please note that this is not a guarantee of publication. Revisions
will be sent through a secondary review process before a final decision is made.

If you would like to make changes to your financial disclosure, please include your
updated statement in your cover letter. Guidelines for resubmitting your figure
files are available below the reviewer comments at the end of this letter.

We look forward to receiving your revised manuscript.

Kind regards,

Melissa Sharer

Academic Editor

PLOS ONE

Journal Requirements:

2. Please include additional information regarding the survey or questionnaire used
in the study and ensure that you have provided sufficient details that others could
replicate the analyses. For instance, if you developed a questionnaire as part of
this study and it is not under a copyright more restrictive than CC-BY, please
include a copy, in both the original language and English, as Supporting
Information, or include a citation if it has been published previously.

3. In the Methods, please discuss whether and how the questionnaire was validated
and/or pre-tested. If these did not occur, please provide the rationale for not
doing so.

4. Please ensure that you refer to Figure 1 in your text as, if accepted, production
will need this reference to link the reader to the figure.

5. We note you have included a table to which you do not refer in the text of your
manuscript. Please ensure that you refer to Table 1 in your text; if accepted,
production will need this reference to link the reader to the Table.

Reviewers' comments:

Reviewer's Responses to Questions

**Comments to the Author**

1. Is the manuscript technically sound, and do the data support the conclusions?

Reviewer #1: Yes

Reviewer #2: Partly

2. Has the statistical analysis been performed
appropriately and rigorously? 

Reviewer #1: Yes

Reviewer #2: N/A

3. Have the authors made all data underlying the
findings in their manuscript fully available?

Reviewer #1: Yes

Reviewer #2: Yes

4. Is the manuscript presented in an intelligible
fashion and written in standard English?

Reviewer #1: Yes

Reviewer #2: Yes

5. Review Comments to the Author

Reviewer #1: The manuscript is well written and the data and findings are presented
in a clear fashion. My primary concerns with the submission is the time period in
which the data is drawn. While the comparison RIC data is for the period immediately
after transition, the data is five years old. There have been significant changes in
the PEPFAR program since the time in which the study is conducted. At a minimum the
authors should be using more recent data. Also, RIC is but one indicator which could
and should be used to assess sustainability. Was there consideration given to other
clinical indicators associated with the cascade, particularly HST data and viral
load/suppression data?

Reviewer #2: PLOS-One_Review

Program Sustainability Post PEPFAR Direct Service Support in the Western Cape, South
Africa

December 2020

This article provides a timely and important inquiry into the sustainability of
donor-supported global health programs. This is an important topic that should have
far greater attention paid to it in the global health literature. There are several
aspects of the paper that require further development, including more extensive
engagement with the global health literature, greater attention to the context of
the Western Cape province, and relative to their proposed structure of program
development, consideration of the role that community-based HIV/AIDS activists have
played in South African HIV/AIDS politics and policy across the institutional levels
of the state.

Thus, while the paper provides an interesting perspective on an important topic, it
requires major revision before it can be considered for publication. As such, my
recommendation for this paper is for it be revised and resubmitted for review.

First and foremost, there is insufficient engagement with influential approaches to
program design in the field of global health. While these approaches have been
primarily framed around patient-centered health outcomes rather than sustainability,
since the authors focus on retention in care (RIC) as the mechanism through which
sustainability is defined, the analysis is focused on similar parameters. With
respect to approaches that the authors should review and engage with, Partners in
Health (PiH) have advocated for an accompaniment approach to program development
that involves extended engagement with across governmental sectors, civil society,
and communities. In addition, Health Alliance International (HAI) has advocated for
‘diagonal’ approaches to global donor funding that involve coordination and
engagement with local governmental and civil society actors and organizations to
increase the impact of donor aid on communities and produce sustainable
interventions. These are but two of many approaches to sustainability for global
health programs that the authors would do well to consider as part of their
analysis. The analysis is also quite shallow relative to the history of
global/international health, which is a bit troubling. In short, the authors are not
contextualizing their findings relative to important developments/histories in the
field, which limits the impact and significance of their findings.

In addition, there is very little attention paid to the contextualization of this
case study. I find the research to be valuable, but there is very little attention
to precisely how and why the Western Cape stands out relative to other regions in
South Africa relative to HIV/AIDS treatment. I have commented on this at length
below and will refer the authors to my input there, but this point also links to
their recommendations for program design, particularly their focus on facility
managers. Taking individual leadership as a key factor without contextualizing the
conditions within which facility managers operate diminishes the potential
contribution. My own experience with facility managers in the Western Cape was that
those who work closely with the community and HIV/AIDS activists also found success,
but these sort of ties and shared governance did not find their place in the
analysis or the program design recommendations the authors provide.

Regarding the parameters for the analysis, I appreciate that the authors have made
the decision to publish their quantitative data separately. However, to exclude the
financial data on sustainability from this paper entirely is quite limiting. This
sort of data would provide the necessary context for understanding why some clinics
were able to undertake the transition better than others, which remains unclear in
the paper. It would also strengthen the comparative value of the paper, as we don’t
know the size of Pepfar programs relative to public health budgets in the Western
Cape, which would seem critical to contextualizing why the province was able to
absorb former Pepfar-funded NGO staff members into the public health system. It may
be the case that other societies that are Pepfar recipients would be able to
undertake similar measures, but we don’t have the necessary information to undertake
these sort of comparative exercises.

While it is likely beyond the scope of this paper, I would encourage the authors to
think more critically about the broader power dynamics within which their research
is situated. In thinking about “transitions” away from global donor funding, it is
not only Pepfar that is noteworthy, but also the Global Fund. In short, this is a
very important topic that they are engaging with, the implications of which are very
much a life and death matter for many around the world, even more so as the COVID-19
pandemic brings with it economic contraction.

Since many recipient countries remain caught in a situation where conditionalities
associated with debt repayment mean that they cannot increase health spending
without commensurate increases in GDP, the looming crisis of “transitions” in global
health funding mean that many societies will be facing declining levels of donor
support along with shrinking health budgets. While a drug-resistant HIV epidemic is
one possible entailment of this, generalized increases in mortality,
under-nutrition, and suffering are also likely.

All of which is to say, I know that I have been quite critical with my review, but I
have done so because I see the potential impact of this line of research and would
like for the paper to reach its full potential. As such, please take my comments and
critique in the constructive spirit in which they have been given, as I believe that
this is an important topic that needs far greater attention.

Comments

Line 62

Retention in Care (RIC) – define acronym with first use

Line 105 – 108

It is true that many countries began to augment transnational donor funding during
this time, but it is also important to note that the 2008 financial crisis produced
a levelling-off in donor funding, and that with access to HIV/AIDS treatment and
small decreases in HIV incidence, that the number of PLHIV continued to increase
during this time, necessitating that recipient governments augment their HIV/AIDS
programs.

Line 108-109

It would also be important to note that the World Bank recategorized country income
levels according to new criteria during this time, which led to restrictions on
donor flows, such as with the delineation of middle-income countries (MICs). That
these criteria are based on aggregate income levels and do not take into account
levels of social inequality has been an important critique of this process.

Line 140

A bit more background on why the donor community defines sustainability along
financial lines is a very important issue to contextualize. From the roots of early
international health programs led by the Pan-American Health Organization (PAHO),
which were funded by the Rockefeller Foundation, to the rise of selective primary
healthcare and cost-recovery amid structural adjustment in the 1980s, there is a
clear thread whereby donors and countries that are able to exert authority within
international institutions express power by defining program sustainability and
impact according to criteria that they set, one of which is financial
sustainability.

Line 150 – 152

Please see James Pfeiffer’s research on Mozambique on how Pepfar-funded interventions
interact with public health systems, as it is more complex than these programs
“working inside” public health clinics, day hospitals, etc.

Line 163-164

This is a good point, but it is also important to note that the emphasis on access to
treatment was also based on the logic of treatment as prevention (TasP), or that
people on HIV/AIDS treatment with undetectable viral loads could not transmit the
virus to other people.

Line 169

On the lack of formal analysis of the transition in Pepfar programmatic leadership:
who was to funded this? Who should have dedicated staff to examine this? There is an
implicit critique here but it is left undefined, leading the reader to assume that
the South African government should have done so. Is this the authors’ position? If
so, please define.

Line 174

A bit more context here on why the Western Cape was distinct is important. The
concentration of tertiary services and expertise is significant, but that is
informed by the uneven historical development of health services around white, urban
populations that began during the colonial era, continued during apartheid, and has
not been resolved during the post-apartheid era. Also, HIV/AIDS programs were
developed earlier in the province under the guidance of Fareed Abdullah and his
team, particularly with PMTCT, which national government intervened to stop during
the dark days of the Mbeki era. But support from the Global Fund in the province,
which came earlier than the rest of the country, set up the Western Cape as an early
success story and a province that has continued to exhibit stronger relative program
management than others. Also, I believe that a similar process had already been
undertaken with a Global Fund grant in the late 2000s, so there would have been
institutional knowledge on how to manage the transition of donor programs from the
government to an NGO.

Line 194

I hope that the financial aspects of the transition are not left out entirely, as
that would weaken the robustness of the analysis considerably.

Line 200

In terms of defining RIC, a bit more would be helpful. I’m assuming that you are
referring to people living with HIV/AIDS (PLHIV) who were adhering to treatment that
were lost to follow up (LTFU)/did not adhere after the transition on Pepfar
programs? Clarifying this would be helpful.

Line 216

Define primary healthcare (PHC) for first use

Line 232

Of what level were the staff nurses? Clinic Nurse Practitioners would be the
assumption, but please define that for the reader.

Line 253

Please define modified grounded theory

Line 274

Defining the donor as an NGO is confusing here, since Pepfar is a donor program
funded by the US gov’t. If you are referring to a primary recipient organization
(PRO) that is working with secondary recipient organization (SRO), then that needs
to be defined. Also, Pepfar, the Global Fund, and several other major global health
programs are public-private partnerships (PPPs), so thinking through how you define
these relationships is important, as simply labeling the donor and NGO limits the
applicability and impact of your findings. Also, is the grantee always local
government? This section needs to be thought through much more.

Line 277

Were there any parameters for defining what the coordinating position would be
enabled to do/oversee? This seems a bit general and undefined since it is the key
point in the section.

Line 280

Donor funded organization? Seems like there is a missing word here

Line 286 - 288

This is an important point that you are making, but it is not sufficiently
contextualized. What you are observing is that the responsibility for a successful
transition in skill transfer is decentralized and falls to the level of a facility
manager. It would lead one to assume that the better capacitated facilities would
therefore be better suited to have successful transfers. Since the areas with
highest HIV prevalence and greatest need for skills transfer (the peri-urban
townships) also tend to have over-burdened heal facilities, this is a critical point
that should be further contextualized.

Line 303

Define NIMART for the reader

Line 314 – 31

Again, this had already been done by the WCDoH previously, with the transition of
clinic management for the HIV/TB clinics established by Médecins sans Frontières
(MSF) in Khayelitsha. There were some bumps in the road with these transitions, and
it was a learning process. So, again, the success here is not surprising but the
result of previous experience in navigating precisely this sort of transition.

Line 331

Again, the fact that the extra labor associated with ensuring a sustainable
transition is being transferred to the facility level is really an important point.
It is not being managed by the donor, provincial health, or city health, but by the
facility manager. There would be a huge range of outcomes to be expected then, which
would depend not only on the personal attributes of the facility manager, but the
extent to which the facility is fully staffed, operational, etc. so that the manager
has the ability to focus on the transition. Also, I would assume that CNPs would
play a critical role in facilitating this transition, as they often have the
clearest understanding of staffing needs, shortages, and areas where increased
efficiencies can be achieved. Their labor, however, is often rendered invisible in
this process, which is problematic.

Line 352

Define IMCI

Line 391

It is helpful that you are addressing the contextual factors here, but this is
insufficient to frame your findings, which are quite particular to the Western
Cape.

Line 396

It would be helpful to mention the role of HIV/AIDS activism in producing this
change, as this was critical to enabling the shifts you identify

Line 400

There is a long-standing debate on the limits of vertical, disease-specific
interventions and their lack of sustainability. Proponents of using “vertical”
interventions to strengthen the broader health system (horizontal) interventions
have advocated for doing precisely what you advise here, to create “diagonal”
programs that use vertical funding streams for health systems strengthening. Health
Alliance International (HAI) has done work that has modeled this approach in
Mozambique is one of the most significant examples of the potential impact and
success of this approach in the global health. In general, I would recommend linking
your case study and discussion of findings to the global health literature, as your
case study is constructed as a stand-alone example, when in reality it is part of a
broader conversation on how best to channel donor funding to improve health
outcomes.

Line 411

Is it because facilities are smaller or that they may be struggling to meet the level
of need in communities with high burdens of disease?

Line 422

Does it make sense to mention the 90-90-90 goals for the first time in the
conclusion? If this is the aim of the paper, then it would make sense to introduce
this goal (which we are projected to miss significantly by the way) earlier in the
paper.

Line 437

Who are national stakeholders? Does they include civil society and PLHIV or HIV/AIDS
activists?

Also, shouldn’t provincial government work with facilities and communities to
understand local needs?

Line 452

What are local champions? Who defines needs? What is the role of the community in
this process?

Line 461

It might be useful to include the provincial treasury in the key stakeholders
meetings, since presumably they will need to plan several years in advance if a
transition will create greater budgetary demands for the health sector. The medium
term expenditure framework (MTEF) requires such advance planning for budgetary
processes.

Line 465

I would include the community or some proxy thereof in the final box in this section.
The role of HIV/AIDS activists as counsellors and mentors who were also consulted by
the WCDoH early on in the development of HIV/AIDS programs was vital to their
success.

Line 471

Who is funding the staffing/capacity required to develop the transition plan? Is this
being done by external consultants?

Line 472

The recommendation to have the skills transfer managed at the provincial level is
contradicted by your evidence, which showed that facility managers oversaw this
process.

Line 474

Given that the entire focus of your paper is to emphasize the importance and lack of
research on how care is affected by a donor transition, I am very surprised that
there is not a post-transition phase for research or monitoring/evaluation. Your
proposed flow of program transition would therefore reproduce the precise issue that
your paper aims to rectify.

6. PLOS authors have the option to publish the peer
review history of their article (what does this mean?). If published, this will
include your full peer review and any attached files.

If you choose “no”, your identity will remain anonymous but your review may still be
made public.

**Do you want your identity to be public for this peer review?** For
information about this choice, including consent withdrawal, please see our
Privacy Policy.

Reviewer #1: No

Reviewer #2: **Yes: **Theodore Powers

---

## [Author Response · Author response to Decision Letter 0]

15 Mar 2021

Dear Plos One,

Please find my responses to the reviewers below. 

I would also like to thank Professor Powers for a thorough and detailed review of my
work. I really appreciate the time you spent reviewing it. 

 Thank you.

 Jessica Chiliza 

2. Please include additional information regarding the survey or questionnaire used
in the study and ensure that you have provided sufficient details that others could
replicate the analyses. For instance, if you developed a questionnaire as part of
this study and it is not under a copyright more restrictive than CC-BY, please
include a copy, in both the original language and English, as Supporting
Information, or include a citation if it has been published previously.

3. In the Methods, please discuss whether and how the questionnaire was validated
and/or pre-tested. If these did not occur, please provide the rationale for not
doing so.

I did not develop a questionnaire, but I validated the interview guide with two
Health Facility Managers. I noted this on lines 294-295. I also have attached the
three different interview guides I used with government leaders, NGO managers and
health facility staff. 

¬¬¬¬¬¬¬¬¬¬¬¬¬¬¬¬¬¬-_______________________________

4. Please ensure that you refer to Figure 1 in your text as, if accepted, production
will need this reference to link the reader to the figure.

Noted and updated. Thank you. 

¬¬¬¬¬¬¬¬¬¬¬¬¬¬¬¬¬¬-_______________________________

5. We note you have included a table to which you do not refer in the text of your
manuscript. Please ensure that you refer to Table 1 in your text; if accepted,
production will need this reference to link the reader to the Table.

Noted and edited. Thank you. 

¬¬¬¬¬¬¬¬¬¬¬¬¬¬¬¬¬¬-_______________________________

Comments to the Author

1. Is the manuscript technically sound, and do the data support the conclusions?

Reviewer #1: Yes

Reviewer #2: Partly

2. Has the statistical analysis been performed appropriately and rigorously?

Reviewer #1: Yes

Reviewer #2: N/A

3. Have the authors made all data underlying the findings in their manuscript fully
available?

Reviewer #1: Yes

Reviewer #2: Yes

4. Is the manuscript presented in an intelligible fashion and written in standard
English?

Reviewer #1: Yes

Reviewer #2: Yes

5. Review Comments to the Author

Reviewer #1: The manuscript is well written and the data and findings are presented
in a clear fashion. My primary concerns with the submission is the time period in
which the data is drawn. While the comparison RIC data is for the period immediately
after transition, the data is five years old. There have been significant changes in
the PEPFAR program since the time in which the study is conducted. At a minimum the
authors should be using more recent data. Also, RIC is but one indicator which could
and should be used to assess sustainability. Was there consideration given to other
clinical indicators associated with the cascade, particularly HST data and viral
load/suppression data?

At the time of this study (2018-2019) the most recent RIC data at a facility level
available from the Western Cape Department of Health (WCDoH) was used. While other
indicators of facility performance were considered the availability and
comprehensive nature of the WCDoH data lent itself to characterizing facility
performance. Clinical data is not easily available at a central source that allows
facility specific aggregation.

The indicators used to measure sustainability needed to be outcomes of the PEPFAR
program. Initially we considered using 4 indicators (proportion of new HIV cases
identified, the proportion of people on treatment, proportion of people who
continued treatment, mortality data and viral loads). When we initially analyzed all
four indicators we realized simplifying to focus on the most sensitive measure of
performance, which was RIC, would allow us to characterize facility performance to
guide the selection of the most widely performing facilities for the interviews.

¬¬¬¬¬¬¬¬¬¬¬¬¬¬¬¬¬¬-_______________________________

Reviewer #2: PLOS-One_Review

Program Sustainability Post PEPFAR Direct Service Support in the Western Cape, South
Africa

December 2020

This article provides a timely and important inquiry into the sustainability of
donor-supported global health programs. This is an important topic that should have
far greater attention paid to it in the global health literature. There are several
aspects of the paper that require further development, including more extensive
engagement with the global health literature, greater attention to the context of
the Western Cape province, and relative to their proposed structure of program
development, consideration of the role that community-based HIV/AIDS activists have
played in South African HIV/AIDS politics and policy across the institutional levels
of the state.

Thus, while the paper provides an interesting perspective on an important topic, it
requires major revision before it can be considered for publication. As such, my
recommendation for this paper is for it be revised and resubmitted for review.

First and foremost, there is insufficient engagement with influential approaches to
program design in the field of global health. While these approaches have been
primarily framed around patient-centered health outcomes rather than sustainability,
since the authors focus on retention in care (RIC) as the mechanism through which
sustainability is defined, the analysis is focused on similar parameters. With
respect to approaches that the authors should review and engage with, Partners in
Health (PiH) have advocated for an accompaniment approach to program development
that involves extended engagement with across governmental sectors, civil society,
and communities. In addition, Health Alliance International (HAI) has advocated for
‘diagonal’ approaches to global donor funding that involve coordination and
engagement with local governmental and civil society actors and organizations to
increase the impact of donor aid on communities and produce sustainable
interventions. 

Thank you for these references. My understanding from the PIH accompaniment approach
is that this was focused on patient adherence to ART, which does not relate to
program sustainability. While these examples clearly provide assistance, improving
continuity of care for HIV defaulters our research was focused on program
sustainability not individual continuity of care. 

(Mukherjee JS, Barry D, Weatherford RD, Desai IK, Farmer PE. Community-Based ART
Programs: Sustaining Adherence and Follow-up. Curr HIV/AIDS Rep. 2016;13(6):359-366.
doi:10.1007/s11904-016-0335-7)

Yes, HAI has advocated for SWAp and various other approaches to coordinating programs
and asking for broader health system support, but vertical programing budgets have
outweighed the Ministry of Health’s attempts at changing these practices. I have
added a section on this the discussion. 

These are but two of many approaches to sustainability for global health programs
that the authors would do well to consider as part of their analysis. The analysis
is also quite shallow relative to the history of global/international health, which
is a bit troubling. In short, the authors are not contextualizing their findings
relative to important developments/histories in the field, which limits the impact
and significance of their findings.

Thank you for this comment. I added a few paragraphs in the Program Sustainability
section. 

In addition, there is very little attention paid to the contextualization of this
case study. I find the research to be valuable, but there is very little attention
to precisely how and why the Western Cape stands out relative to other regions in
South Africa relative to HIV/AIDS treatment. 

I added a paragraph about how the Western Cape is different from the rest of South
Africa. Please refer to lines 209-216.

I have commented on this at length below and will refer the authors to my input
there, but this point also links to their recommendations for program design,
particularly their focus on facility managers. Taking individual leadership as a key
factor without contextualizing the conditions within which facility managers operate
diminishes the potential contribution. My own experience with facility managers in
the Western Cape was that those who work closely with the community and HIV/AIDS
activists also found success, but these sort of ties and shared governance did not
find their place in the analysis or the program design recommendations the authors
provide.

I regret that I did not formally enquire about the ties between activism and
closeness with the community. I found some facility managers were skeptical of HIV
activists and others were HIV activists themselves. 

Regarding the parameters for the analysis, I appreciate that the authors have made
the decision to publish their quantitative data separately. However, to exclude the
financial data on sustainability from this paper entirely is quite limiting. This
sort of data would provide the necessary context for understanding why some clinics
were able to undertake the transition better than others, which remains unclear in
the paper. It would also strengthen the comparative value of the paper, as we don’t
know the size of Pepfar programs relative to public health budgets in the Western
Cape, which would seem critical to contextualizing why the province was able to
absorb former Pepfar-funded NGO staff members into the public health system. It may
be the case that other societies that are Pepfar recipients would be able to
undertake similar measures, but we don’t have the necessary information to undertake
these sort of comparative exercises.

I completely agree with you. In my original proposal we included the financial data
for individual NGO’s and different health facility levels but when we started
collecting information it was impossible and would have totally changed the thesis
and for that reason left out. It remains a valuable suggestion. I hope someone who
has an accounting background would take this up. It is surprising it has not been
conducted by PEPFAR. It would be invaluable. We thought we could do it but the focus
was not on financial sustainability but on what makes for sustainable outcomes. 

While it is likely beyond the scope of this paper, I would encourage the authors to
think more critically about the broader power dynamics within which their research
is situated. In thinking about “transitions” away from global donor funding, it is
not only Pepfar that is noteworthy, but also the Global Fund. In short, this is a
very important topic that they are engaging with, the implications of which are very
much a life and death matter for many around the world, even more so as the COVID-19
pandemic brings with it economic contraction.

Since many recipient countries remain caught in a situation where conditionalities
associated with debt repayment mean that they cannot increase health spending
without commensurate increases in GDP, the looming crisis of “transitions” in global
health funding mean that many societies will be facing declining levels of donor
support along with shrinking health budgets. While a drug-resistant HIV epidemic is
one possible entailment of this, generalized increases in mortality,
under-nutrition, and suffering are also likely.

All of which is to say, I know that I have been quite critical with my review, but I
have done so because I see the potential impact of this line of research and would
like for the paper to reach its full potential. As such, please take my comments and
critique in the constructive spirit in which they have been given, as I believe that
this is an important topic that needs far greater attention.

Comments

Line 62

Retention in Care (RIC) – define acronym with first use

Noted and edited.

Line 105 – 108

It is true that many countries began to augment transnational donor funding during
this time, but it is also important to note that the 2008 financial crisis produced
a levelling-off in donor funding, and that with access to HIV/AIDS treatment and
small decreases in HIV incidence, that the number of PLHIV continued to increase
during this time, necessitating that recipient governments augment their HIV/AIDS
programs.

Noted and edited.

Line 108-109

It would also be important to note that the World Bank recategorized country income
levels according to new criteria during this time, which led to restrictions on
donor flows, such as with the delineation of middle-income countries (MICs). That
these criteria are based on aggregate income levels and do not take into account
levels of social inequality has been an important critique of this process.

Noted and edited.

Line 140

A bit more background on why the donor community defines sustainability along
financial lines is a very important issue to contextualize. From the roots of early
international health programs led by the Pan-American Health Organization (PAHO),
which were funded by the Rockefeller Foundation, to the rise of selective primary
healthcare and cost-recovery amid structural adjustment in the 1980s, there is a
clear thread whereby donors and countries that are able to exert authority within
international institutions express power by defining program sustainability and
impact according to criteria that they set, one of which is financial
sustainability.

Thank you for this comment. I added a section which provides more context. 

Line 150 – 152

Please see James Pfeiffer’s research on Mozambique on how Pepfar-funded interventions
interact with public health systems, as it is more complex than these programs
“working inside” public health clinics, day hospitals, etc.

Thank you for this comment. I added a paragraph which explains PEPFAR’s influence on
the health system. 

¬¬¬¬¬¬¬¬¬¬¬¬¬¬¬¬¬¬-_______________________________

Line 163-164

This is a good point, but it is also important to note that the emphasis on access to
treatment was also based on the logic of treatment as prevention (TasP), or that
people on HIV/AIDS treatment with undetectable viral loads could not transmit the
virus to other people.

This concept came later. While this was the rationale for the TB programs there was
limited evidence until much later and used in the PEPFAR program.

¬¬¬¬¬¬¬¬¬¬¬¬¬¬¬¬¬¬¬_______________________________

Line 169

On the lack of formal analysis of the transition in Pepfar programmatic leadership:
who was to funded this? Who should have dedicated staff to examine this? There is an
implicit critique here but it is left undefined, leading the reader to assume that
the South African government should have done so. Is this the authors’ position? If
so, please define.

This is a good point that is addressed in the Conclusion section.

Line 174

A bit more context here on why the Western Cape was distinct is important. The
concentration of tertiary services and expertise is significant, but that is
informed by the uneven historical development of health services around white, urban
populations that began during the colonial era, continued during apartheid, and has
not been resolved during the post-apartheid era. Also, HIV/AIDS programs were
developed earlier in the province under the guidance of Fareed Abdullah and his
team, particularly with PMTCT, which national government intervened to stop during
the dark days of the Mbeki era. But support from the Global Fund in the province,
which came earlier than the rest of the country, set up the Western Cape as an early
success story and a province that has continued to exhibit stronger relative program
management than others. Also, I believe that a similar process had already been
undertaken with a Global Fund grant in the late 2000s, so there would have been
institutional knowledge on how to manage the transition of donor programs from the
government to an NGO.

In your last sentence I assume you mean transition from NGO to government?

I consulted with the WCDoH, who said there was never a formal transfer of Global Fund
human resources or programming over to government. Global Funded human resources
applied for WCDoH posts when they were advertised. 

Line 194

I hope that the financial aspects of the transition are not left out entirely, as
that would weaken the robustness of the analysis considerably.

We analyzed PEPFAR expenditure figures for South Africa from 2007-2015.

While this allowed us to see the dip in funding from 2012-2014 in 2015 PEPFAR funding
increased again when the Global AIDS Coordinator decided to fund direct service
again. 

Figure 9: PEPFAR and SAG Expenditure HIV and TB

2007-2015 (ZAR)*

Source: 

2007-2010: South African Consolidated HIV and TB Spending Assessment
2007/8-2009/10

2011-2014: South African HIV and TB Investment Case, Reference Report

2014-2016: Consolidated HIV and TB Spending Assessment 2014/15-2016/17

¬¬¬¬¬¬¬¬¬¬¬¬¬¬¬¬¬¬-_______________________________

Line 200

In terms of defining RIC, a bit more would be helpful. I’m assuming that you are
referring to people living with HIV/AIDS (PLHIV) who were adhering to treatment that
were lost to follow up (LTFU)/did not adhere after the transition on Pepfar
programs? Clarifying this would be helpful.

The definition of RIC used in this study is also used by the WCDoH. RIC was
calculated per health facility per year among adults (age >15),

First line + Second line + Third Line + Clients stopped ART / (Total on treatment –
Total transferred out)

“Total on treatment” includes the HIV clients who transferred into the health
facility, via a formal or silent transfer. Silent transfers were considered new ART
initiates, in the absence of a patient tracking system. Mortality dropped out of the
RIC calculation. A decision was made based on a sensitivity analysis that an
alternative RIC definition would not significantly change the outcome. 

¬¬¬¬¬¬¬¬¬¬¬¬¬¬¬¬¬¬-_______________________________

Line 216

Define primary healthcare (PHC) for first use

I edited PHC. Thank you.

¬¬¬¬¬¬¬¬¬¬¬¬¬¬¬¬¬¬-_______________________________

Line 232

Of what level were the staff nurses? Clinic Nurse Practitioners would be the
assumption, but please define that for the reader.

The level of nurse was 1 staff nurse and 1 clinical nurse practitioner. I noted these
distinctions in the paper. See lines 287-289.

¬¬¬¬¬¬¬¬¬¬¬¬¬¬¬¬¬¬-_______________________________

Line 253

Please define modified grounded theory

This was also a misunderstanding on my part. It should just be grounded theory. 

¬¬¬¬¬¬¬¬¬¬¬¬¬¬¬¬¬¬-_______________________________

Line 274

Defining the donor as an NGO is confusing here, since Pepfar is a donor program
funded by the US gov’t. If you are referring to a primary recipient organization
(PRO) that is working with secondary recipient organization (SRO), then that needs
to be defined. Also, Pepfar, the Global Fund, and several other major global health
programs are public-private partnerships (PPPs), so thinking through how you define
these relationships is important, as simply labeling the donor and NGO limits the
applicability and impact of your findings. Also, is the grantee always local
government? This section needs to be thought through much more.

Thank you for this comment. I edited my paper referring to either “local government,
NGO or donor (ie.PEPFAR).

¬¬¬¬¬¬¬¬¬¬¬¬¬¬¬¬¬¬_______________________________________¬¬¬¬¬¬¬¬¬¬¬¬¬¬¬¬¬¬-_______________________________

Line 277

Were there any parameters for defining what the coordinating position would be
enabled to do/oversee? This seems a bit general and undefined since it is the key
point in the section.

This person would be responsible for ensuring the transparency of donor funded
activities and work with government to ensure the program is integrated into the
local health system. I clarified this in the paper. 

¬¬¬¬¬¬¬¬¬¬¬¬¬¬¬¬¬¬-_______________________________

Line 280

Donor funded organization? Seems like there is a missing word here

This has been edited.

¬¬¬¬¬¬¬¬¬¬¬¬¬¬¬¬¬¬-_______________________________

Line 286 - 288

This is an important point that you are making, but it is not sufficiently
contextualized. What you are observing is that the responsibility for a successful
transition in skill transfer is decentralized and falls to the level of a facility
manager. It would lead one to assume that the better capacitated facilities would
therefore be better suited to have successful transfers. Since the areas with
highest HIV prevalence and greatest need for skills transfer (the peri-urban
townships) also tend to have over-burdened heal facilities, this is a critical point
that should be further contextualized.

To clarify, my point is that the skills transfer needs to both centralized at a
provincial level and decentralized at a facility level. The provincial level is
needed so that essential donor funded staff are transferred to the public health
system, which needs coordination in terms of adequate budgeting. At a health
facility level, Health Facility Managers should ensure that donor funded staff who
are being let go train local health facility staff in their job
responsibilities.

I clarified this point in the paper. 

¬¬¬¬¬¬¬¬¬¬¬¬¬¬¬¬¬¬-_______________________________

Line 303

Define NIMART for the reader

Edited. Thank you

¬¬¬¬¬¬¬¬¬¬¬¬¬¬¬¬¬¬-_______________________________

Line 314 – 31

Again, this had already been done by the WCDoH previously, with the transition of
clinic management for the HIV/TB clinics established by Médecins sans Frontières
(MSF) in Khayelitsha. There were some bumps in the road with these transitions, and
it was a learning process. So, again, the success here is not surprising but the
result of previous experience in navigating precisely this sort of transition.

My understanding is the MSF work in Khayelitsha Site B for the past 20 years. They
have worked in the same health facility. I know that they have conducted research on
transitioning patients into adult ART treatment programs. I am aware of their work
on adherence clubs, which was later adapted and scaled up throughout the province. 

The WCDoH did not have experience transitioning programs and human resources as vast
as PEPFAR. PEPFAR was very different in that they also were not transparent with
their work so the WCDoH was not aware of the vast amount of programming in the
province. PEPFAR was supporting 435 human resources in the Western Cape in 2012. So
yes, the WCDoH may have had some experience transitioning some patients into the
health system, but it was not at the scale of the PEPFAR experience. 

¬¬¬¬¬¬¬¬¬¬¬¬¬¬¬¬¬¬-_______________________________

Line 331

Again, the fact that the extra labor associated with ensuring a sustainable
transition is being transferred to the facility level is really an important point.
It is not being managed by the donor, provincial health, or city health, but by the
facility manager. There would be a huge range of outcomes to be expected then, which
would depend not only on the personal attributes of the facility manager, but the
extent to which the facility is fully staffed, operational, etc. so that the manager
has the ability to focus on the transition. Also, I would assume that CNPs would
play a critical role in facilitating this transition, as they often have the
clearest understanding of staffing needs, shortages, and areas where increased
efficiencies can be achieved. Their labor, however, is often rendered invisible in
this process, which is problematic.

These are all very good points and are highlighted in the Conclusion section.

¬¬¬¬¬¬¬¬¬¬¬¬¬¬¬¬¬¬¬_______________________________¬¬¬¬¬_________________

Line 352

Define IMCI

Noted and edited. 

¬¬¬¬¬¬¬¬¬¬¬¬¬¬¬¬¬¬-_______________________________

Line 391

It is helpful that you are addressing the contextual factors here, but this is
insufficient to frame your findings, which are quite particular to the Western
Cape.

 Point taken this will be noted in the conclusion
¬¬¬¬¬¬¬¬¬¬¬¬¬¬¬¬¬¬¬_______________________________

Line 396

It would be helpful to mention the role of HIV/AIDS activism in producing this
change, as this was critical to enabling the shifts you identify.

Good point added to sentence about political support.

¬¬¬¬¬¬¬¬¬¬¬¬¬¬¬¬¬¬-_______________________________

Line 400

There is a long-standing debate on the limits of vertical, disease-specific
interventions and their lack of sustainability. Proponents of using “vertical”
interventions to strengthen the broader health system (horizontal) interventions
have advocated for doing precisely what you advise here, to create “diagonal”
programs that use vertical funding streams for health systems strengthening. Health
Alliance International (HAI) has done work that has modeled this approach in
Mozambique is one of the most significant examples of the potential impact and
success of this approach in the global health. In general, I would recommend linking
your case study and discussion of findings to the global health literature, as your
case study is constructed as a stand-alone example, when in reality it is part of a
broader conversation on how best to channel donor funding to improve health
outcomes.

As the reviewer states this is a long running debate and will be addressed again in
the conclusion

Line 411

Is it because facilities are smaller or that they may be struggling to meet the level
of need in communities with high burdens of disease?

The reason for this is usually smaller health facilities with few staff are not able
to let health staff just focus on one disease. So the donor funded HIV staff will
usually get absorbed into the facilities and end up being a generalist, attending to
all patients. This point was added to the conclusion. 

¬¬¬¬¬¬¬¬¬¬¬¬¬¬¬¬¬¬-_______________________________

Line 422

Does it make sense to mention the 90-90-90 goals for the first time in the
conclusion? If this is the aim of the paper, then it would make sense to introduce
this goal (which we are projected to miss significantly by the way) earlier in the
paper.

I added a section at the beginning (line 260-264) regarding the 90-90-90 goals. 

¬¬¬¬¬¬¬¬¬¬¬¬¬¬¬¬¬¬¬_______________________________

Line 437

Who are national stakeholders? Does they include civil society and PLHIV or HIV/AIDS
activists?

Also, shouldn’t provincial government work with facilities and communities to
understand local needs?

I defined these points in the paper. 

¬¬¬¬¬¬¬¬¬¬¬¬¬¬¬¬¬¬-_______________________________

Line 452

What are local champions? Who defines needs? What is the role of the community in
this process?

I defined these points in the paper in Table 4 . 

¬¬¬¬¬¬¬¬¬¬¬¬¬¬¬¬¬¬¬_______________________________

Line 461

It might be useful to include the provincial treasury in the key stakeholders
meetings, since presumably they will need to plan several years in advance if a
transition will create greater budgetary demands for the health sector. The medium
term expenditure framework (MTEF) requires such advance planning for budgetary
processes.

I defined these points in the paper (see Table 5). 

¬¬¬¬¬¬¬¬¬¬¬¬¬¬¬¬¬¬-_______________________________

Line 465

I would include the community or some proxy thereof in the final box in this section.
The role of HIV/AIDS activists as counsellors and mentors who were also consulted by
the WCDoH early on in the development of HIV/AIDS programs was vital to their
success.

Thank you. I added this detail to the paper. 

¬¬¬¬¬¬¬¬¬¬¬¬¬¬¬¬¬¬-_______________________________

Line 471

Who is funding the staffing/capacity required to develop the transition plan? Is this
being done by external consultants?

Thank you for this comment. I clarified this point in the paper. 

¬¬¬¬¬¬¬¬¬¬¬¬¬¬¬¬¬¬-_______________________________

Line 472

The recommendation to have the skills transfer managed at the provincial level is
contradicted by your evidence, which showed that facility managers oversaw this
process.

This is not what I was conveying. Please see my comment from lines 286 – 288
above

Line 474

Given that the entire focus of your paper is to emphasize the importance and lack of
research on how care is affected by a donor transition, I am very surprised that
there is not a post-transition phase for research or monitoring/evaluation. Your
proposed flow of program transition would therefore reproduce the precise issue that
your paper aims to rectify.

This is a very good point. I have added a section that includes the post-transition
time period (see Table 8).

6. PLOS authors have the option to publish the peer review history of their article
(what does this mean?). If published, this will include your full peer review and
any attached files.

If you choose “no”, your identity will remain anonymous but your review may still be
made public.

Do you want your identity to be public for this peer review? For information about
this choice, including consent withdrawal, please see our Privacy Policy.

Reviewer #1: No

Reviewer #2: Yes: Theodore Powers

---

## [Decision Letter · Decision Letter 1]

7 Apr 2021

PONE-D-20-37215R1

Program Sustainability Post PEPFAR Direct Service Support in the

Western Cape, South Africa

PLOS ONE

Dear Jessica Chiliza,

Thank you for submitting your manuscript to PLOS ONE. After careful consideration, we
feel your submission has been strengthened and requires minor revision.  Therefore,
we invite you to submit a revised version of the manuscript that addresses the
points raised during the review process.  We appreciate your efforts to deepen our
community's understanding of the process of sustainability as it aligns with
external/PEPFAR funding in the context of South Africa.

If you would like to make changes to your financial disclosure, please include your
updated statement in your cover letter. Guidelines for resubmitting your figure
files are available below the reviewer comments at the end of this letter.

We look forward to receiving your revised manuscript.

Kind regards,

Melissa Sharer

Academic Editor

PLOS ONE

Journal Requirements:

Additional Editor Comments (if provided):

 Reviewers' comments:

Reviewer's Responses to Questions

**Comments to the Author**

1. If the authors have adequately addressed your comments raised in a previous round
of review and you feel that this manuscript is now acceptable for publication, you
may indicate that here to bypass the “Comments to the Author” section, enter your
conflict of interest statement in the “Confidential to Editor” section, and submit
your "Accept" recommendation.

Reviewer #1: (No Response)

Reviewer #2: All comments have been addressed

2. Is the manuscript technically sound, and do the data
support the conclusions?

Reviewer #1: No

Reviewer #2: Yes

3. Has the statistical analysis been performed
appropriately and rigorously? 

Reviewer #1: N/A

Reviewer #2: N/A

4. Have the authors made all data underlying the
findings in their manuscript fully available?

Reviewer #1: Yes

Reviewer #2: Yes

5. Is the manuscript presented in an intelligible
fashion and written in standard English?

Reviewer #1: Yes

Reviewer #2: Yes

6. Review Comments to the Author

Reviewer #1: N/A 

Reviewer #2: The authors have significantly strengthened the paper through revision,
and my recommendation is that the article be accepted for publication pending minor
revisions, which are detailed below.

Line 75

This study suggests

Line 101

In 2008, or following 2008?

Line 111

Missing word after “increased”

Line 119

PEPFAR misspelled. Also, “after PEPFAR withdrawal” or “after the withdrawal of PEPFAR
funding”

Line 122

Insert “and reductions in” prior to “tracing systems”

Line 171

It looks like there is an incomplete sentence here with “The donor community
capacity”

Paragraph starting on Line 197

This is an excellent addition to the argument.

Section starting on Line 203

This is very helpful for situating the particularity of the Western Cape.

Paragraph starting on Line 253

It might be useful to signal that the 90-90-90 rhetoric emanating from the UN has
coincided with the leveling off of donor funding and the “transition” processes
initiated by PEPFAR and the Global Fund. In short, while we have the tools to “end
HIV/AIDS”, funding that would have otherwise supported this approach has been
withdrawn, which has undermined this program. Perhaps this would fit best in the
conclusion, but it would be worth mentioning, as this critical dynamic seems to be
lost on many.

Line 350

Consider rewording to: “understand the context, and local policy, and have…”

Line 351

Consider rewording to: “to the context, which builds trust and results in more”

Line 356

Consider rewording to: “and at a decentralized level.”

Line 450

Consider rewording to: “centralized level: either the provincial or district
level.”

Line 455

Consider rewording to: “important for government and the NGO to be”

Line 467

This interview excerpt has already been used (Line 419). Please delete one of these
so that there is not repetition.

Line 491

Consider rewording to: “the main concern is that if the public-sector workforce and
infrastructure are undermined”

Line 511

Start new paragraph with: “Donors”

Line 546

Consider rewording to: “need to be intentionally”

Line 554

Under “Grantee”: “Ideally have an Establish a donor coordination”

Line 577

Perhaps consider including a wider array of inputs on the transition planning
process. Certainly, while the donors will appreciate the objectivity of an external
consultant, this is a critical moment in ensuring the long-term sustainability of
the program, and there should therefore be a clear and thorough consultation process
that involves the full range of stakeholders.

---

## [Author Response · Author response to Decision Letter 1]

16 Apr 2021

Responses to Reviewers #2

Dear Plos One,

Please find my responses in Bold to the reviewers. 

 Thank you.

 Jessica Chiliza 

Journal Requirements:

Reference #39 was added to include the decrease in global HIV funding as 90-90-90
targets were implemented. 

Additional Editor Comments (if provided):

 Reviewers' comments:

Reviewer's Responses to Questions

Comments to the Author

1. If the authors have adequately addressed your comments raised in a previous round
of review and you feel that this manuscript is now acceptable for publication, you
may indicate that here to bypass the “Comments to the Author” section, enter your
conflict of interest statement in the “Confidential to Editor” section, and submit
your "Accept" recommendation.

Reviewer #1: (No Response)

Reviewer #2: All comments have been addressed

2. Is the manuscript technically sound, and do the data support the conclusions?

Reviewer #1: No

Reviewer #2: Yes

3. Has the statistical analysis been performed appropriately and rigorously? 

Reviewer #1: N/A

Reviewer #2: N/A

4. Have the authors made all data underlying the findings in their manuscript fully
available?

Reviewer #1: Yes

Reviewer #2: Yes

5. Is the manuscript presented in an intelligible fashion and written in standard
English?

Reviewer #1: Yes

Reviewer #2: Yes

6. Review Comments to the Author

Reviewer #1: N/A 

Reviewer #2: The authors have significantly strengthened the paper through revision,
and my recommendation is that the article be accepted for publication pending minor
revisions, which are detailed below.

Line 75

This study suggests

Noted and edited. 

Line 101

In 2008, or following 2008?

Noted and edited. 

Line 111

Missing word after “increased”

Noted and edited. 

Line 119

PEPFAR misspelled. Also, “after PEPFAR withdrawal” or “after the withdrawal of PEPFAR
funding”

Noted and edited. 

Line 122

Insert “and reductions in” prior to “tracing systems”

Noted and edited.

Line 171

It looks like there is an incomplete sentence here with “The donor community
capacity”

Noted. It looks part of my sentence was missing. Thank you. 

Paragraph starting on Line 197

This is an excellent addition to the argument.

Thank you.

Section starting on Line 203

This is very helpful for situating the particularity of the Western Cape.

Thank you. 

Paragraph starting on Line 253

It might be useful to signal that the 90-90-90 rhetoric emanating from the UN has
coincided with the leveling off of donor funding and the “transition” processes
initiated by PEPFAR and the Global Fund. In short, while we have the tools to “end
HIV/AIDS”, funding that would have otherwise supported this approach has been
withdrawn, which has undermined this program. Perhaps this would fit best in the
conclusion, but it would be worth mentioning, as this critical dynamic seems to be
lost on many.

Thank you. I added a sentence on line 253 and in the conclusion with regard to this
decrease in funding. 

Line 350

Consider rewording to: “understand the context, and local policy, and have…”

Noted and edited.

Line 351

Consider rewording to: “to the context, which builds trust and results in more”

Thank you. I edited the whole sentence (lines 350 and 351) to the following:

“When an NGO has an established office in the geographic region, they understand the
context, local policy and have strong relationships with government , which builds
trust and results in more sustainable outcomes.”

Line 356

Consider rewording to: “and at a decentralized level.”

Noted and edited.

Line 450

Consider rewording to: “centralized level: either the provincial or district
level.”

Noted and edited.

Line 455

Consider rewording to: “important for government and the NGO to be”

Noted and edited.

Line 467

This interview excerpt has already been used (Line 419). Please delete one of these
so that there is not repetition.

Oh my! Thank you very much for picking this up. Noted and Edited. 

Line 491

Consider rewording to: “the main concern is that if the public-sector workforce and
infrastructure are undermined.”

Noted and edited.

Line 511

Start new paragraph with: “Donors”

Noted and edited.

Line 546

Consider rewording to: “need to be intentionally”

Noted and edited.

Line 554

Under “Grantee”: “Ideally have an Establish a donor coordination”

Edited. Thank you!

Line 577

Perhaps consider including a wider array of inputs on the transition planning
process. Certainly, while the donors will appreciate the objectivity of an external
consultant, this is a critical moment in ensuring the long-term sustainability of
the program, and there should therefore be a clear and thorough consultation process
that involves the full range of stakeholders.

Thank you for this comment. I have added a few additional inputs. 

to Reviewers #2.docx
---

## [Editor Report · Decision Letter 2]

23 Apr 2021

Program Sustainability Post PEPFAR Direct Service Support in the

Western Cape, South Africa

PONE-D-20-37215R2

Dear Dr. Chiliza,

We’re pleased to inform you that your manuscript has been judged scientifically
suitable for publication and will be formally accepted for publication once it meets
all outstanding technical requirements.

Kind regards,

Melissa Sharer, PhD MPH MSW

Academic Editor

PLOS ONE
---

## [Editor Report · Acceptance letter]

14 May 2021

PONE-D-20-37215R2 

Program Sustainability Post PEPFAR Direct Service Support in the Western Cape, South
Africa 

Dear Dr. Chiliza:

I'm pleased to inform you that your manuscript has been deemed suitable for
publication in PLOS ONE. Congratulations! Your manuscript is now with our production
department. 

Kind regards, 

on behalf of

Dr. Melissa Sharer 

Academic Editor

PLOS ONE